# *Microcystis viridis NIES-102* Cyanobacteria Lectin (MVL) Interacts with SARS-CoV-2 Spike Protein Receptor Binding Domains (RBDs) via Protein–Protein Interaction

**DOI:** 10.3390/ijms25126696

**Published:** 2024-06-18

**Authors:** Zhengguang Wang, Zhihan Yang, Mami Shishido, Khadija Daoudi, Masafumi Hidaka, Hiroaki Tateno, Eugene Futai, Tomohisa Ogawa

**Affiliations:** 1Laboratory of Enzymology, Graduate School of Agricultural Sciences, Tohoku University, Sendai 980-8572, Japan; wang.zhengguang.s3@dc.tohoku.ac.jp (Z.W.); yang.zhihan.s2@dc.tohoku.ac.jp (Z.Y.); mami.shishido.p2@dc.tohoku.ac.jp (M.S.); daoudi.khadija.b8@tohoku.ac.jp (K.D.); masafumi.hidaka.a4@tohoku.ac.jp (M.H.); eugene.futai.e1@tohoku.ac.jp (E.F.); 2Cellular and Molecular Biotechnology Research Institute, National Institute of Advanced Industrial Science and Technology, Tsukuba 305-8566, Japan; h-tateno@aist.go.jp

**Keywords:** lectin, SARS-CoV-2, cyanobacteria, carbohydrate specificity, protein–protein interaction

## Abstract

The emergence of coronavirus disease 2019 (COVID-19) posed a major challenge to healthcare systems worldwide, especially as mutations in the culprit Severe Acute Respiratory Syndrome Coronavirus 2 (SARS-CoV-2) complicated the development of vaccines and antiviral drugs. Therefore, the search for natural products with broad anti-SARS-CoV-2 capabilities is an important option for the prevention and treatment of similar infectious diseases. Lectins, which are widely recognized as antiviral agents, could contribute to the development of anti-SARS-CoV-2 drugs. This study evaluated the binding affinity of six lectins (including the cyanobacterial lectin from *Microcystis viridis* NIES-102 (MVL), and Jacalin, a lectin from the breadfruit, *Artocarpus altilis*) to the receptor binding domain (RBD) of the spike protein on the original (wild) SARS-CoV-2 and three of its mutants: Alpha, Delta, and Omicron. MVL and Jacalin showed distinct binding affinity to the RBDs of the four SARS-CoV-2 strains. The remaining four lectins (DB1, ConA, PHA-M and CSL3) showed no such binding affinity. Although the glycan specificities of MVL and Jacalin were different, they showed the same affinity for the spike protein RBDs of the four SARS-CoV-2 strains, in the order of effectiveness Alpha > Delta > original > Omicron. The verification of glycan-specific inhibition revealed that both lectins bind to RBDs by glycan-specific recognition, but, in addition, MVL binds to RBDs through protein–protein interactions.

## 1. Introduction

Enveloped viruses, characterized by a lipid bilayer derived from the host cell membrane, represent a diverse group of pathogens [1]. In particular, those such as human immunodeficiency virus (HIV) and coronavirus (CoV) pose a significant threat to humans. The process of entry of the enveloped virus genome into the target cell involves two primary steps: initial envelope binding to cell surface receptors and subsequent fusion with the cell membrane [2,3,4]. As a type of CoV, Severe Acute Respiratory Syndrome (SARS)-Cov-2 represents a class of RNA coronaviruses characterized by an enveloped structure, with a diameter of approximately 80–160 nm [5]. The enveloped structure of SARS-CoV-2 is composed of a lipid membrane with three types of glycoproteins on its surface: the spike protein (S), which functions in receptor recognition, cell lysis, and primary antigen; the small envelope glycoprotein (E), which facilitates attachment to the host cell membrane; and the membrane glycoprotein (M), which is responsible for the formation of the envelope and the budding of new viruses [6,7,8].

The virus achieves its invasion primarily by binding to host cell receptors with the S protein [9,10]. The S protein contains two subunits, S1 and S2, and forms a cloverleaf-shaped trimer with three S1 heads and a trimeric S2 stalk. S1 can be further subdivided into an N-terminal domain (NTD), receptor binding domain (RBD) and C-terminal domain (CTD) [11]. The RBD is located at the tip of the S1 head [12]. The S1 subunit of SARS-CoV-2 is primarily responsible for recognizing and binding to the angiotensin-converting enzyme 2 (ACE2) on the host cell, facilitating the virus’s entry into the host cell [13]. The RBD is also the main source of antigen variation and immune escape, such as, for example, the SARS-CoV-2 Omicron mutant (now with a worldwide distribution), which has 17 amino acid mutations in the RBD region but also presents a target for the development of preventive anti-SARS-CoV-2 vaccines and drugs [14]. The high variability of RNA viruses frequently results in the ineffectiveness of vaccines or acquisition of drug resistance. To address such issues, it is crucial to find antiviral compounds that act through different mechanisms, such as compounds that target differentially glycosylated binding of the RBD, or those that undergo protein–protein interactions with conserved domains of the RBD.

Lectins are a diverse group of proteins that bind carbohydrates and agglutinate erythrocytes [15,16]. Many lectins with antiviral properties have been identified, particularly those that target the densely glycosylated RBD of enveloped viruses. The *Oscillatoria agardhii* (Cyanobacteria) agglutinin homologue (OAAH) lectin family exhibits robust anti-influenza virus activity through specific binding to high-mannose glycans present on the envelope glycoprotein hemagglutinin, effectively preventing viral entry into host cells [17]. Griffithsin (GRFT), a lectin isolated from the red alga *Griffithsia*, shows remarkable broad-spectrum antiviral activity. Previous studies have demonstrated the ability of GRFT to inhibit HIV infection at picomolar concentrations, as well as the ability to induce resistance against the SARS and Middle East Respiratory Syndrome (MERS) viruses [18,19,20]. The cyanovirin-N (CV-N) lectin, in addition to its inhibitory effect on HIV, also has the potential to treat the filoviral Ebola disease through saccharide-specific recognition [21]. Undoubtedly, cyanobacterial and algal lectins have a vast potential as natural antiviral compounds, suggesting promising applications in various fields. *Microcystis viridis* lectin (MVL) was isolated from the unicellular freshwater bloom-forming cyanobacterium *M. viridis* NIES-102 strain [22]. MVL consists of 113 amino acid residues, including two tandemly repeated homologous 54-residue carbohydrate binding domains [22], a homodimer stabilized by a boomerang-shaped monomer domain interlock [23]. MVL also inhibits HIV-1 envelope-mediated cell fusion at nanomolar concentrations by binding to high-mannose N-linked carbohydrate on the surface of the envelope glycoprotein gp120 [24].

The present study focuses on screening and the mechanistic exploration of MVL with anti-SARS-CoV-2 potential against four SARS-CoV-2 variants. Recombinant MVL was prepared and assessed for its binding properties against spike protein RBDs of the SARS-CoV-2 variants in comparison with other lectins: Jacalin, which is a galactose-specific lectin [25]; a mannose-specific lectin from the Chinese yam, *Dioscoria batata* (DB1) [26]; Concanavalin A (Con A) from the jackbean *Canavalia ensiformis* [27]; phytohemagglutinin M-form (PHA-M), a lectin from the legume *Phaseolus vulgaris* [28]; and CSL3, an egg lectin from the chum salmon, *Oncorhynchus keta* [29].

## 2. Results

### 2.1. Expression, Purification and Characterization of Recombinant MVL

Recombinant MVL (rMVL) was successfully expressed as a thioredoxin (TrxA) fusion protein using a pET32a vector (Figure 1a and Appendix A). Although optimization attempts included the use of several different types of host cell, and different culture temperature and conditions, rMVL-TrxA was expressed in the inclusion bodies under all tested conditions (Appendix A).

To obtain soluble rMVL, purification was conducted under denaturing conditions, with subsequent refolding of rMVL-TrxA. Briefly, the inclusion bodies were denatured in 8 M urea and then purified by nickel affinity chromatography (Appendix A). Subsequently, the rMVL-TrxA was refolded by dialysis at 4 °C with 20 mM Tris buffer (pH 7.8) to remove the urea. The TrxA tag (13.9 kD) was then removed by thrombin digestion, and purified rMVL (15.6 kDa) was obtained (Figure 1b and Appendix A). The activity of rMVL and other lectins was confirmed by hemagglutination assays with a 2% solution of rabbit erythrocytes. The minimum hemagglutinating concentration of rMVL was 0.008 mg.mL^−1^, showing a slightly lower activity than the reported native MVL (0.0015 mg.mL^−1^: Table 1 and Appendix A).

### 2.2. Interaction and Binding Properties of rMVL with SARS-CoV-2 Spike Protein RBDs

The binding properties of six lectins, of which structural properties (L-type for Con A and PHA-M, M-type for DB1, MVL, Jacalin-related and SUEL-related) and carbohydrate binding specificities are diverse (Table 1), against SARS-CoV-2 spike protein RBDs of four variants were evaluated by surface plasmon resonance (SPR) technology (Figure 2). MVL and Jacalin showed spike protein RBD binding affinity with all SARS-CoV-2 variants (Figure 2), while the other lectins showed no or very weak binding affinity (Table 2).

Table 2 summarizes the binding properties of these lectins. In particular, the galactose-specific lectin Jacalin interacted effectively with the spike protein RBDs of all four variants, (Alpha, Delta, Omicron and original) with similar *K*_D_ values ranging from 3.20 to 4.78 × 10^−7^ M (Figure 2 and Table 2). In contrast, the mannose-specific lectin MVL [23] also showed interactions with all variants, but its binding affinity was slightly lower than that of Jacalin, with *K*_D_ values ranging from 1.29 to 2.73 × 10^−6^ M (Figure 2 and Table 2). The order (highest to lowest) of binding affinity for MVL with each SARS-CoV-2 spike protein RBD was in the order Alpha, Delta, original and Omicron RBDs. The other mannose-specific lectins, Con A, PHA-M, DB1 and CSL3, did not interact with the spike protein RBD. On the other hand, the difference in affinity between each mutant and the lectins also demonstrates that the mutations in RBDs among the four SARS-CoV-2 strains affect the binding ability of lectins. In particular, the affinity between Omicron, which has the most mutant sites, and the lectins was lowest for both rMVL and Jacalin lectins.

### 2.3. Carbohydrate Binding Specificity of MVL

MVLs are considered to be mannose binding lectins [23], but their detailed glycan specificities are not known. Therefore, the glycan specificity of rMVL was analyzed using glycan array analysis technology in comparison with mannose binding DB1. As shown in Figure 3, rMVL recognized 13 types of glycans in 4 categories, including complex-type, desialylated complex-type, agalactosylated complex-type and high-mannose-type N-glycans. In contrast, DB1 recognized five types of glycans in three categories, including complex-type, desialylated complex-type, and high-mannose-type glycans.

rMVL binds to four glycoproteins, Fetuin, α1-acid glycoprotein, transferrin and thyroglobulin, each of which has a different saccharide terminus (Figure 3a,c). rMVL showed the highest binding to glycans containing desialylated complex-type N-glycans with a galactose terminus (No. 43–46 in Figure 3a,c). The highest binding was observed for asialo transferrin at a relative fluorescence unit (RFU) of 26,218 (No. 45 in Figure 3a,c) and asialo porcine thyroglobulin at 20,746 RFU (No. 46 in Figure 3a,c). The rMVL binding affinity to glycans containing complex-type N-glycans and agalactosylated complex-type N-glycans was lower than to that of desialylated complex-type N-glycans (Nos. 25–28 or 49–51 in comparison with Nos. 43–46 in Figure 3a,c). This suggests that the galactose terminus enhances the binding. Therefore, the binding to glycans including complex-type N-glycans varied among glycoproteins, with RFUs ranging from 2882 (α1-acid glycoprotein) to 18,224 (porcine thyroglobulin glycan), suggesting relatively low binding to the NeuAc terminus. Conversely, binding to glycans containing agalactosylated complex-type N-glycans did not exceed 1000 RFUs, indicating relatively low binding to the GlcNAc terminus. The binding effects of MVL lectin to high-mannose-type N-glycans were also low: 1577 RFUs (ovalbumin) and 3370 RFUs (invertase). These results suggest that rMVL preferentially binds to the galactose terminus in comparison with the mannose, GlcNAc and NeuAc termini.

DB1 lectin exhibited recognition exclusively towards five tested glycans sharing the saccharide chain of high-mannose-type N-glycan (Nos. 28, 46, 53, 57 and 94 in Figure 3b,d), confirming that DB1 lectin recognizes the mannose terminus. This lectin showed the highest binding affinity of 26,503 RFUs with the yeast invertase glycan, and the lowest binding affinity of 476 RFUs with the hybrid-type N-glycan of ovoalbumin (Nos. 57 and 53 in Figure 3b,d). These results indicate that MVL exhibits saccharide binding specificity distinct from that of DB1 and suggests that this specificity is related to their different interactions with the spike protein RBDs. Consequently, it is concluded that rMVL is a mannose/galactose-specific binding lectin rather than just a mannose binding lectin.

### 2.4. Galactose Inhibits the Interaction between Lectins and RBDs

The inhibitory effects of specific glycans on lectin binding showed that untreated Jacalin bound well to the Alpha and Omicron spike protein RBDs. However, after galactose treatment (30 min), Jacalin binding was inhibited almost completely in a concentration-dependent manner, even at the lowest concentration of galactose (90% inhibition at 12.5 mM for Alpha and 80% at 12.5 mM for Omicron; Figure 4c,d, 100% inhibition at 12.5 mM for Delta and 60% at 12.5 mM for original; Appendix A), suggesting that Jacalin and spike protein RBDs interact via glycan-specific recognition.

The effect of galactose on the MVL was different: untreated MVL could bind to Alpha and Omicron spike protein RBDs, although galactose inhibition of MVL binding occurred (as with Jacalin), treatment of MVL with 100 mM galactose did not completely inhibit the interaction of MVL with the Alpha or Omicron spike protein RBDs, and approximately 40% of the SPR signal was retained. Furthermore, there was no significant difference between the response at the lowest galactose concentration of 12.5 mM and the highest concentration of 100 mM (Figure 4a,b). These results indicate that the interaction between MVL and spike protein RBDs is only partially dependent on galactose-specific interactions. These results were also observed for Delta and original SARS-CoV-2 RBDs (Appendix A). When the carbohydrate recognition sites of MVL were blocked with excessive amounts of galactose (1000-times more), nearly half of the MVL retained binding affinity to RBDs. We therefore propose that the interaction between MVL lectin and spike protein RBDs involves not only glycan-specific recognition, but also protein–protein interaction.

### 2.5. The Protein–Protein Interaction between MVL and RBDs

To explore the protein–protein interaction between MVL and RBDs, an in silico method was used to predict the binding sites and modes between them. A structural comparison of RBD mutants predicted by AlphaFold2 (Figure 5a) suggests that their structures are almost identical, even though the Omicron RBD contains 16 mutation sites from the original, while other RBDs (Alpha and Delta) have 1 or 2 amino acid substitutions.

Subsequently, docking analysis (using HDOCK) between the four spike protein RBDs and MVL was performed, and the complex structure with the lowest docking energy score was selected, which were as follows: Alpha, −293.37 kcal/mol; Delta, −296.37 kcal/mol; Omicron, −307.74 kcal/mol; and original, −294.01 kcal/mol (Figure 6).

The protein–protein interaction sites of Alpha, Delta and original RBDs with MVL showed a similar docking energy and common amino acid residues; that is, the six residues of these RBDs, including Y421, Y449, R457, Y473, A475 and Q493, were involved in the interaction with MVL (Figure 6). In addition, in the Alpha and Delta RBDs, Y489 was involved in the interaction with MVL, and Y501, a mutated amino acid residue in Alpha, was also involved in the interaction between Alpha RBD and MVL. On the other hand, the Omicron variant, which has 16 RBD mutations compared to the original virus, was different from other variants (Figure 6c). The Omicron RBD has only two amino acid residues bound to MVL, Y473 and N417. Although the Y473 residue is present in the other three RBDs, N417 was the unique RBD mutation amino acid site of the Omicron variant (Figure 6c). One remarkable feature in the binding site of RBDs to MVL is that tyrosine was the most predominant, occupying almost half of the position.

However, it should be noted that the lowest energy score with the highest confidence showed that MVL bound to the C-terminal amino acids of RBDs, contributing to binding to the ACE2 receptor (Figure 5b). The docking results of the second to tenth lowest energy scores showed that MVL bound to RBDs in a different manner (Appendix A). These results suggest that MVL interacts with SARS-CoV-2 S-protein RBDs at the binding site of RBDs to ACE2 through protein–protein interaction.

## 3. Discussion

In this study, the interaction of six lectins with four SARS-CoV-2 spike protein RBDs was analyzed using SPR. Two galactose-specific lectins, MVL and Jacalin, showed high affinity to SARS-CoV-2 RBDs, although they have no structural similarities such as motifs or families. Many mannose binding lectins have been reported to recognize highly mannosylated SARS-CoV-2 RBDs [31]; however, the mannose binding lectins analyzed in this study, ConA, PHA-M and DB1, did not interact with the RBDs. The most significant finding in this study is that MVL can bind to the spike protein RBD through protein–protein interaction. When saccharide-chain recognition was blocked by galactose, Jacalin binding to the SARS-CoV-2 RBD was lost, but MVL was still able to bind to the spike protein RBDs, suggesting a protein–protein interaction.

In silico docking analysis suggested the presence of protein–protein interaction between MVL and spike protein RBDs. The complex structure models showed that MVL interacted with SARS-CoV-2 spike protein RBDs mainly through polar contacts. Lan et al. performed X-ray crystallography analysis with a complex composed of recombinant SARS-CoV-2 RBD and ACE2 expressed in Hi5 insect cells [30]. Their analysis showed that a total of 17 RBD residues were in contact with 20 residues of ACE2 and that these 17 residues were mainly located at the C-terminal end of the RBD amino acid sequence, from K417 to Y505 (Figure 5b) [31]. In silico docking results demonstrated that the binding sites of the four RBDs to MVL were also located at the C-terminal end from amino acid residue 417, and three of them (Y449, Y489, and Q493) were associated with the amino acid residue position of RBD binding to hACE2. The other binding sites of RBDs to MVL, although not identical, were also in the vicinity of the binding site of RBD and hACE2. These findings imply that MVL prevents the binding of RBD to hACE2 by masking the amino acid binding site between RBD and hACE2. It is worth noting that many π − π stacks in MVL were also involved to varying degrees in binding with RBDs. The π − π stack refers to the attractive, non-covalent interactions between aromatic rings, which are strongly associated with the broad-spectrum antiviral properties of lectins [32,33]. At the same time, tyrosine was most predominant in the binding site of RBDs to MVL, which may also be related to the π − π stack. The amino acids phenylalanine, tyrosine and tryptophan are significantly involved in π interactions, such as π − π stack and H-π bonds [34]. According to the in silico results, Tyr residues of RBDs are actively involved in the π − π stack, such as Y449-W38 and Y489-Y4, and other Tyrs contribute to the interaction of lectins with RBDs in the form of H-π interactions. This non-covalent interaction between hydrogen atoms and aromatic rings also provides the basis for the protein–protein interaction between MVL lectin and RBDs.

The number of sites of polar interaction of the four mutant RBDs with MVL was eight for Alpha, seven for Delta, six for original, and two for Omicron. These numbers of interactions were consistent with the order of affinity between MVL and each mutant RBD, as measured by SPR; that is, more binding sites produce higher affinity than fewer.

In addition, MVL can recognize complex-type N-glycans, desialylated complex-type N-glycans, and agalactosylated complex-type N-glycans, all of which end in galactose. It has been reported that the RBD expressed in HEK293 cells has galactose groups in the terminal N-glycan chains at residues N331 and N343 [34,35]. These glycosylation properties of spike protein RBDs were thought to be the main reason why Jacalin and MVL, but not other lectins such as Con A and DB1, bind to spike protein RBDs. The lectins reported to recognize the spike protein or RBD of SARS-CoV-2 are mostly high-mannose binding lectins such as GRFT lectin derived from the red marine alga *Griffithsia* sp. and BanLec lectin isolated from the banana *Musa acuminata*. Both high-mannose lectins can effectively inhibit the binding of SARS-CoV-2 spike protein to the hACE2 receptor at nM concentrations [36,37]. The SARS-CoV-2 spike protein contains a total of 22 glycosylation sites, including high-mannose sites, hybrid sites and complex-type N-glycosylation sites, with high-mannose-type glycosylation sites accounting for the majority [38]. Therefore, high-mannose-type lectins, GRFT and BanLec can inhibit SARS-CoV-2 infection by recognizing the high-mannose glycosylation sites and binding to the S1 or S2 region of the SARS-CoV-2 spike protein. However, few studies have also reported the interaction between some lectins that specifically recognize the complex-type N-glycan with a galactose moiety and the SARS-CoV-2 RBD. The FRIL isolated from the edible lablab or hyacinth bean, *Lablab purpureus*, is a complex-type N-glycan-binding lectin that can recognize glycans with the outermost galactose group and simultaneously has a weak affinity for mannose, similar to MVL. FRIL can effectively recognize SARS-CoV-2 galactose glycoprotein and interfere with SARS-CoV-2 entry into cells [39].

MVL showed strong binding activity against the RBD of SARS-CoV-2 spike proteins, while its saccharide specificity is galactose rather than the previously reported mannose [22]; that is, MVL bound more strongly to asialo complex-type N-glycan (galactose) rather than high-mannose-type glycans (mannose) (Figure 3). This may be the reason why MVL can bind well to RBD in comparison with other mannose lectins. Furthermore, MVL showed strong binding activity to the RBD of SARS-CoV-2 spike protein, even in the presence of galactose, suggesting a sugar chain-independent interaction. In silico analysis suggested that this sugar chain-independent interaction between MVL and RBD is a protein–protein interaction. Therefore, to explain this difference in mannose binding lectins, it will be necessary to investigate further by analyzing the interaction between MVL and RBDs at the atomic level using cryo-electron microscopy.

Research on lectin action on virus envelopes has focused mainly on lectin binding to virus RBD through carbohydrate-specific recognition and inhibition of virus invasion [40,41,42]. There has been limited research on the lectin inhibition of binding of a virus RBD to its receptor through protein–protein interaction, except for the calcium-dependent (C-type) lectin fold [43]. However, the discovery here that MVL has dual antiviral potency, through both protein–protein interaction and carbohydrate-specific recognition, warrants further research on the antiviral interactions and mechanisms of such lectins.

## 4. Materials and Methods

### 4.1. Materials

PHA-M (Cat# L-8902) and ConA (Cat# C-2010) lectins were purchased from Sigma-Aldrich Japan Ltd. (Tokyo, Japan). Jacalin lectin (Cat# L-1150) was purchased from Vector Laboratories Inc. (Newark, CA, USA). The lectins were obtained as lyophilized powders, stored at −30 °C, and during use, they were dissolved in 100 mM Tris-HCl buffer at a concentration of 1 mg.mL^−1^. DB1 lectin was purified as reported previously [26]. Briefly, the homogenized sample from *Dioscorea batatas* yam tubers was first purified by ammonium sulfate precipitation, followed by hydrophobic interaction chromatography on a Phenyl-Toyopearl 650 M column, with final purification by anion-exchange chromatography on a HiTrap Q column with a NaCl gradient. CSL3 was purified from chum salmon (*Oncorhynchus keta*) eggs, as previously described [29], and stored at −30 °C until use. The anti-EGFP used for Western blot detection was purchased from Fujifilm Wako (Osaka, Japan); anti-mouse IgG secondary antibody was obtained from Roche Diagnostics K.K. (Tokyo, Japan). All other reagents were of the purest grade commercially available.

### 4.2. Expression and Purification of Recombinant MVL

For the expression of recombinant *Microcystis viridis* NIES-102 lectin (rMVL), cDNA encoding MVL was subcloned into the pVT118N vector, as reported previously [22]. After introducing the restriction enzyme sites, NcoI and HindIII, by polymerase chain reaction (PCR) using MVL forward and reverse primers, the PCR fragment of MVL was introduced into the pET32a vector by ligation with T4 DNA ligase; then, the resulting plasmids were transformed into DH5α-competent cells. The following day, a single positive colony was seeded into LB medium (with 0.1% ampicillin) and cultured at 37 °C overnight, following which plasmids were extracted using an extraction kit from Favorgen (Ping Tung, Taiwan). The successfully constructed plasmid was confirmed by sequence analysis and temporarily stored at −30 °C.

The constructed plasmids were transformed into BL21 (DE3) *Escherichia coli*-competent cells and selected by LB plates supplemented with ampicillin. The empty vector pET32a was also transformed into BL21 (DE3) as a negative control. A single colony was inoculated into 3.5 mL of liquid LB (with 0.1% ampicillin) and pre-cultured at 37 °C overnight. Then, the bacteria were inoculated in 200 mL LB (0.1% ampicillin) at a ratio of 1:100 and incubated with shaking at 37 °C until OD_600_ reached 0.4–0.6. After approaching the target concentration, 0.5 mM of IPTG was added to induce expression at 23 °C for a further 4 h. Subsequently, the *E. coli* bacteria were harvested by centrifugation at 8000× *g* for 10 min, the supernatant was removed, and the precipitate was stored at −30 °C.

The precipitate was resuspended in sonication buffer (50 mM Tris, 100 mM NaCl, pH 8.0) and sonicated on ice for 30 min (30 s on/30 s off). The debris was removed by centrifugation at 18,000× *g* for 15 min. The supernatant and insoluble precipitate were analyzed by 12.5% SDS–PAGE. The inclusion bodies, existing in a precipitated form, were initially washed three times with a cleaning buffer to remove cellular debris and then treated with wash buffer (50 mM Tris, 100 mM NaCl, 10 mM EDTA, 0.5% (*w*/*w*) TritonX-100, pH 8.0) at 4 °C, each time washing for 30 min followed by centrifugation at 15,000× *g* for 10 min. Subsequently, the inclusion bodies were subjected to overnight treatment at 4 °C with denaturation buffer (50 mM Tris, 100 mM NaCl, 1 mM 2-mercaptoethanol, 8 M Urea, pH 7.8). The following day, the supernatant was recovered, after centrifugation at 15,000× *g* for 10 min, and purified by nickel affinity chromatography (ÄKTA Pure Protein Purification System) under denaturation conditions with loading buffer (50 mM Tris, 100 mM NaCl, 1 mM 2-Mercaptoethanol, 5 mM Imidazole, 8 M Urea, pH 7.8). The sample was eluted with imidazole at 20 mM, 100 mM and 250 mM, and fractions were immediately analyzed by 12.5% SDS–PAGE.

Fractions containing the target protein were mixed and dialyzed into thrombin cleavage buffer (20 mM Tris, 30 mM NaCl, 2 mM, CaCl_2_, pH 8.0). Then, 10U thrombin was added to the sample to remove the TrxA-tag at room temperature (RT) in a 2 h reaction terminated by adding PSMF until 2 mM final concentration. Following this, MVL and the TrxA-tag were separated through further nickel affinity chromatography. Pure lectin was obtained at flow through (FT), and TrxA tag was recovered in the 100 mM imidazole fraction. The digested lectin-TrxA samples, FT fraction and 100 mM imidazole fraction were analyzed by SDS-PAGE.

### 4.3. Construction of the SARS-CoV-2 Mutant Spike Protein RBD Expression System

This experiment involved four types of spike protein RBDs of SARS-CoV-2: the original (wild-type) and the three mutants, Alpha, Delta and Omicron. The original SARS-CoV-2 spike protein RBD plasmid, pcDNA3-SARS-CoV-2-S-RBD-sfGFP, was kindly provided by Prof. Erik Procko via Addgene (Watertown, MA, USA) [44]. Using the quick-change method, the Alpha and Delta variants were constructed by site-directed mutagenesis: on the N501Y single mutation site for the Alpha variant; and on the L452R and T478K mutation sites for the Delta variant.

For the Omicron variant, due to numerous mutation sites in its RBD, the In-fusion method was employed to ligate the synthetic Omicron spike protein RBD gene fragment (Integrated DNA Tachnology k.k., Tokyo, Japan) with the pcDNA3-sfGFP vector fragment, to construct the pcDNA3-Omicron-RBD-sfGFP plasmid. Briefly, primers were designed for amplification of the Omicron RBD and pcDNA3-sfGFP vector fragment, then amplified through PCR. The resulting products were identified and recovered by 1.2% agarose gel electrophoresis. Subsequently, the insert fragment and linearized vector fragment were ligated by 5× In-Fusion Snap Assembly Master Mix. Because of the selection resistance of the plasmid to ampicillin, the amplification and extraction methods for the constructed SARS-CoV-2 mutant spike protein RBD plasmid were the same as those for the lectin plasmid construction method described above.

The successfully constructed four RBD plasmids were transfected into HEK293T cells for secretion expression using polyethylenimine (PEI). Briefly, HEK293T cells were thawed and seeded in 10 cm tissue culture dishes with DMEM containing 10% fetal bovine serum (FBS). The cells were cultured for 3 days at 37 °C with 5% CO_2_. The confluence of cells before transfection was about 90%. Each plasmid (10 μg) was mixed well with 250 μL of Opi-EME medium, followed by 25 μL of PEI, and reacted at RT for 15 min. The resulting mixture was transferred to HEK293T cells and incubated for 24 h. The next day, the culture medium was replaced with an FBS-free DMEM, and the cells were cultured for a further 72 h before harvesting.

The harvested culture medium was purified through nickel column affinity chromatography by loading onto a 1 mL pre-packed nickel column using binding buffer (50 mM Tris, 100 mM NaCl, pH 8.0). The elution buffer was composed of 50 mM Tris, 100 mM NaCl, 500 mM Imidazole, pH 8.0. Fractions were collected at 20 mM, 50 mM and 250 mM imidazole concentrations, then subjected to Western blotting analysis (Appendix A).

### 4.4. SDS-PAGE and Western Blotting

To evaluate their expression and purity, the purified MVL fractions were subsequently analyzed using 12.5% SDS-PAGE and stained with Coomassie Brilliant Blue for 1 h and then destained with acetic acid–methanol solution.

The spike protein RBDs were also separated using SDS-PAGE and then transferred onto Polyvinylidene-Fluoride membranes. The membranes were blocked with 5% skimmed milk powder for 1 h at RT and then incubated with anti-EGFP primary antibody at 4 °C overnight. The membranes were then subjected to secondary incubation with a HRP-conjugated anti-mouse IgG secondary antibody, followed by photographic detection. The blots were developed using an ECL system and the intensities of the bands were quantified using an LAS-4000 image analyzer (Fujifilm, Japan).

### 4.5. Hemagglutination Assays

The biological activity of commercial lectin, recombinant lectin and naturally extracted lectin was evaluated based on their aggregation effect on 2% rabbit erythrocytes (Japan Bio Serum Co., Hiroshima, Japan). The experiment was conducted in a U-bottom 96-well plate. Initially, 50 μL of PBS was dispensed into each well, followed by the addition of 50 μL lectin into the wells of the first column. After thorough mixing, the mixture was subjected to two-fold serial dilution into the subsequent columns. The 96-well plate was left to stand at RT for 30 min and observed for hemagglutination. Subsequently, the minimum concentration of each lectin required to induce hemagglutination was determined.

### 4.6. Surface Plasmon Resonance (SPR) Detection of the Interaction between Lectins and SARS-CoV-2 Spike Protein RBDs

The binding affinity (K_D_) of 6 kinds of lectins and 4 kinds of spike protein RBDs was measured at RT by a Biocore T200 molecular interaction instrument. Firstly, the spike protein RBDs and the lectin samples were dialyzed into HBS-EP+ buffer (0.01 M HEPES, 0.15 M NaCl, 3 mM EDTA, 0.05% surfactant P 20, pH 7.4). Then, the spike protein RBDs were diluted to 100 μg/mL in 10 mM sodium acetate (pH 4.5) and immobilized on a CM5 chip after biotinylation according to the instrument manual, reaching a target level of 200 response units (RU), using the Amine Coupling Kit from GE HealthCare (Chicago, IL, USA). As well as a sample flow cell, a reference flow cell (which does not capture any molecular protein) was used to correct protein response contributions, such as bulk shifts. The lectin samples to be analyzed were diluted from 5 μM to 0.31 μM with HBS-EP buffer and passed through the sensor chip at a flow rate of 30 μL.min^−1^ for 120 s. To correct baseline drift or other disturbances, two blank cycles were performed before injecting the lectin samples into the sensor chip. Between sample injections, the sensor chip was regenerated with pH 2.5 glycine at 30 μL.min^−1^ for 60 s to remove any bound analytes and to clean the chip surface for the next analysis cycle. The entire experiment was conducted at 25 °C. After determination, the two-state reaction model of the Biacore T200 evaluation software ver. 1.0 was used for interaction analysis, with the Rmax analysis parameter set to ‘Local’.

The interaction between lectins and spike protein RBDs was analyzed using SPR to determine if it depends on carbohydrate-specific recognition binding or protein–protein interaction binding. The specific carbohydrates appropriate for each lectin were prepared with HBS-EP + buffer. They were then mixed with the lectin samples and allowed to react for 30 min at RT. The final concentration of lectin was 5 μM, and the final concentrations of carbohydrate were 0 mM, 12.5 mM, 25 mM, 50 mM and 100 mM.

### 4.7. Carbohydrate Specificity Analysis of MVL and DB1 Lectins

MVL (10 μg) and DB1 (10 μg) were labeled with Cy3-N-hydroxysuccinimide ester (GE Healthcare), and excess Cy3 was removed with Sephadex G-25 desalting columns (GE Healthcare). Cy3-labeled lectin was diluted with probing buffer (25 mM Tris-HCl, pH 7.5, 140 mM NaCl, 2.7 mM KCl, 1 mM CaCl_2_, 1 mM MnCl_2_ and 1% Triton X-100) to 10 μg.mL^−1^ and incubated with the glycan microarray at 20 °C overnight. The glycan microarray was washed three times with probing buffer, and fluorescence images were captured using a Bio-Rex Scan 200 evanescent-field-activated fluorescence scanner (Rexxam Co. Ltd., Kagawa, Japan).

### 4.8. In Silico Analysis

The spike protein RBD structure of original SARS-CoV-2 was obtained from the RDB database under the ID 6w41 and used as a template to predict the three-dimensional structure of RBDs of Alpha, Delta and Omicron mutants through SWISS-MODEL (https://swissmodel.expasy.org/ (accessed on 12 March 2024)). The three-dimensional structural modeling of MVL was predicted by AlphaFold2 (https://colab.research.google.com/github/sokrypton/ColabFold/blob/main/AlphaFold2.ipynb (accessed on 18/May/2022)). The HDOCK molecular docking web server (http://hdock.phys.hust.edu.cn/ (accessed on 15/May/2024)) was used for visual analysis to analyze the protein–protein interaction surface characteristics between MVL and various RBDs. The docking result with the lowest energy score was selected, and PyMOL software ver. 3.0 was used for visual analysis to analyze the protein–protein interaction surface characteristics between MVL and various RBDs.

## 5. Conclusions

In this study, the anti-SARS-CoV-2 inhibiting ability of six lectins from different sources was screened, and two lectins, Jacalin and MVL, with broad-spectrum anti-SARS-CoV-2 activity were successfully verified. The affinity order of these lectins to four types of SARS-CoV-2 spike protein RBDs was identical: Alpha > Delta > original > Omicron. Jacalin and SARS-CoV-2 RBD interacted via galactose-specific recognition, while MVL and SARS-CoV-2 RBD interacted through both galactose-specific recognition and protein–protein interaction. According to the results of the in silico analysis, MVL mainly bound to SARS-CoV-2 RBD through polar contact via the RBD N-terminal. The number of amino-acid binding sites between MVL and RBD is (from most to least) Alpha > Delta > original > Omicron, which is consistent with the affinity order determined by SPR. In addition, it is considered that π − π stacks may also play a role in the antiviral effect of MVL.

## Figures and Tables

**Figure 1 ijms-25-06696-f001:**
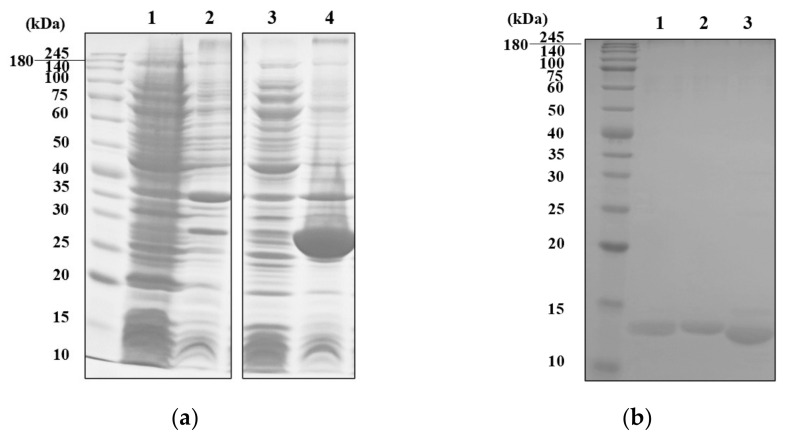
SDS-PAGE profiles of (**a**) rMVL-TrxA expressed by pET32a vector and (**b**) purified rMVL after removal of the TrxA tag. (**a**) Supernatants and pellets of bacterial lysates: supernatant (lanes 1, 3), pellet (lanes 2, 4); from cells with pET32a vector (lanes 1, 2) or pET32a-MVL (lanes 3, 4). (**b**) After digestion of rMVL-TrxA by α-thrombin, rMVL was recovered by nickel affinity. The digested sample with α-thrombin (lane 1), rMVL in flowthrough (lane 2), and the TrxA tag eluted with 250 mM imidazole (lane 3).

**Figure 2 ijms-25-06696-f002:**
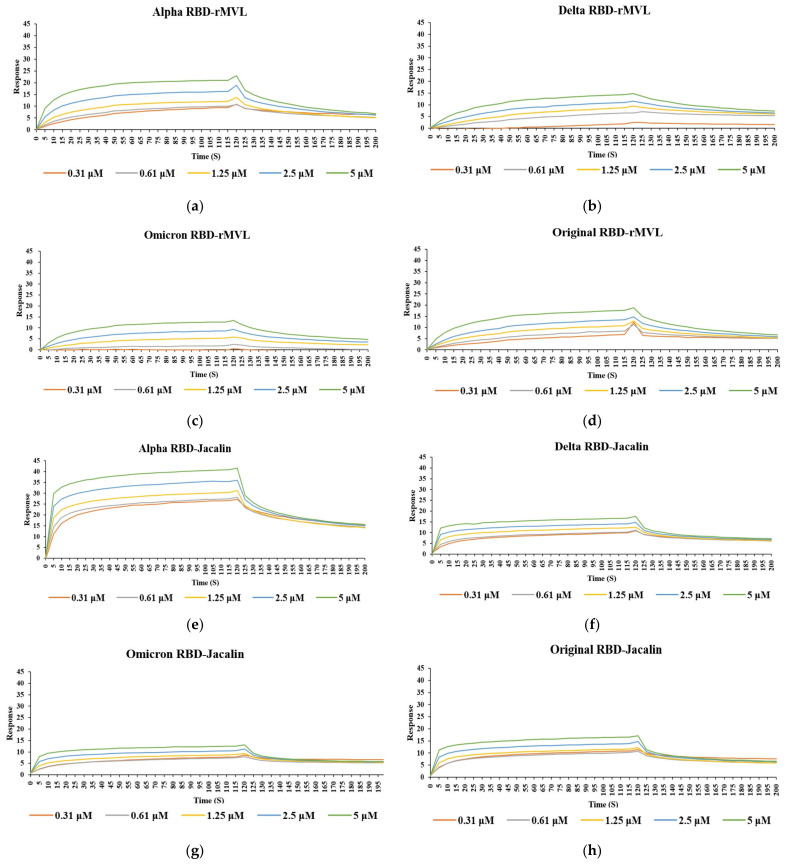
SPR profiles of the interaction between spike protein RBDs and lectins rMVL and Jacalin. Spike protein RBDs of SARS-CoV-2 strains alpha (**a**,**e**), delta (**b**,**f**), omicron (**c**,**g**), and original (**d**,**h**), were used as ligands attached on a CM5 sensor chip in a Biacore T200. rMVL (**a**–**d**) and Jacalin (**e**–**h**) samples were used as analytes at different concentrations (0.31, 0.62, 1.25, 2.5 and 5 µM; traces in orange, gray, yellow, blue, and green, respectively) to record responses to determine the interaction between RBDs and lectins.

**Figure 3 ijms-25-06696-f003:**
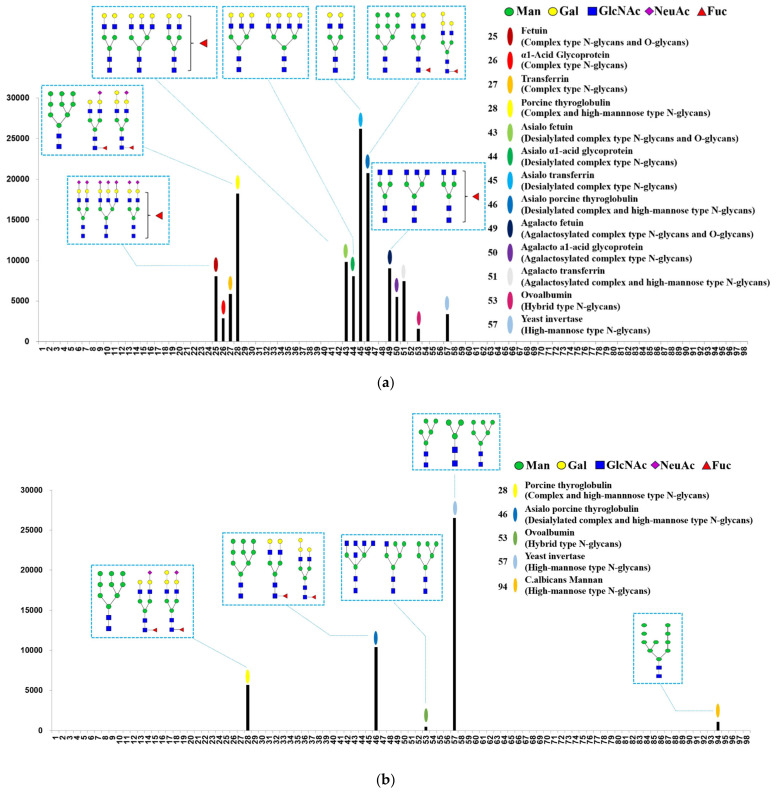
Glycan array analysis profiles of recombinant MVL (**a**) and DB1 (**b**) lectins and lists of the glycans that bind to MVL (**c**) and DB1 (**d**). Symbol nomenclature for glycans (SNFG) is used to represent oligosaccharides on the graph (green circles, mannose; yellow circles, galactose; blue squares, GlcNAc; purple diamonds, NeuAc; red triangles, fucose).

**Figure 4 ijms-25-06696-f004:**
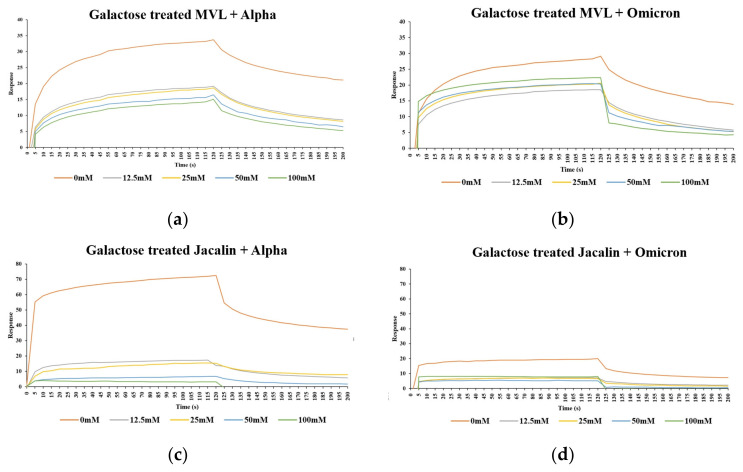
Interaction between lectins and Alpha or Omicron spike protein RBDs after galactose treatment. MVL (**a**,**b**) and Jacalin (**c**,**d**) were treated with galactose (0 mM, 12.5 mM, 25 mM, 50 mM, 100 mM) at room temperature for 30 min before SPR, which was performed as for Figure 2.

**Figure 5 ijms-25-06696-f005:**
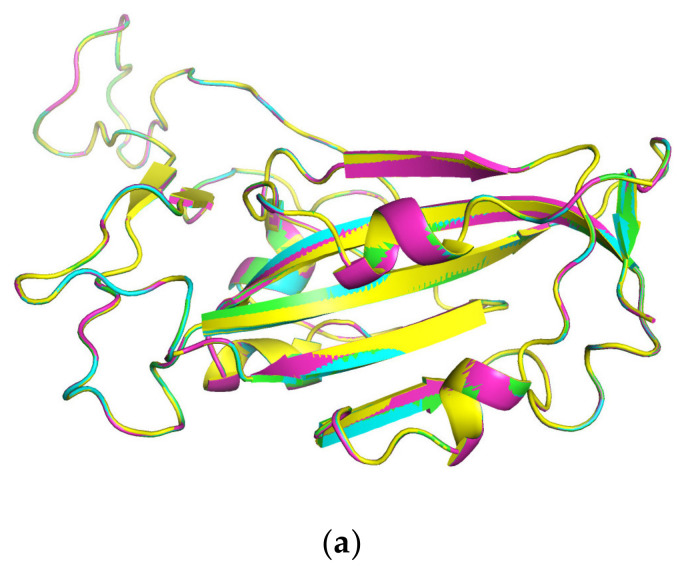
Comparison of the AlphaFold2-predicted three-dimensional structure of four SARS-CoV-2 Alpha, Delta, Omicron and original spike protein RBDs (in blue, magenta, yellow, and light green, respectively) (**a**) and the RBD (brown)-ACE2 (dark green) complex of SARS-CoV-2 (PDB ID 6M0J) (**b**) [30].

**Figure 6 ijms-25-06696-f006:**
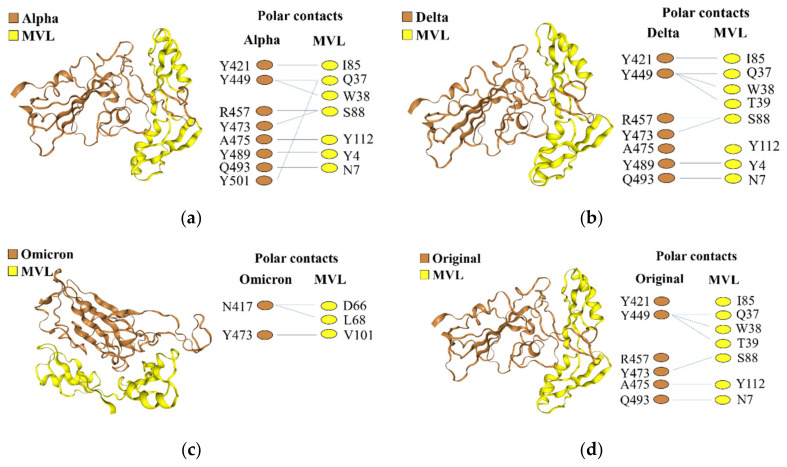
Results of MVL (yellow) and spike protein RBD (brown) docking analysis using HDOCK. (**a**) Alpha RBD, (**b**) Delta RBD, (**c**) Omicron RBD, (**d**) original RBD. Residues showing polar interactions are listed to the right.

**Table 1 ijms-25-06696-t001:** Hemagglutination activity of recombinant and natural lectins.

Lectin Name	Minimum Agglutination Concentration (mg.mL^−1^)	Protein Fold/ Family	Carbohydrate Specificity
rMVL	0.008	α/β-fold/MVL	D-mannose/D-galactose [23], this study
Jacalin	0.005	β-prism/Jacalin-related	D-galactose/GalNAc [25]
Con A	0.008	β-sandwich/L-type	D-mannose [27]
PHA-M	0.002	β-sandwich/L-type	complex-type *N*-glycans [28]
DB1	0.03	β-prism II/M-type	D-mannose [26]
CSL3	0.0002	α/β-fold/SUEL-related	L-rhamnose [29]

**Table 2 ijms-25-06696-t002:** SPR binding kinetics of lectins to the spike protein RBDs of SARS-CoV-2 variants.

Name	Interaction with RBDs	Affinity with RBD [*K*_D_ (M)]			
		Alpha	Delta	Omicron	Original
MVL	Yes	1.29 × 10^−6^	1.65 × 10^−6^	2.73 × 10^−6^	2.08 × 10^−6^
Jacalin	Yes	3.20 × 10^−7^	3.39 × 10^−7^	4.78 × 10^−7^	4.28 × 10^−7^
DB1	No	―	―	―	―
Con.A	No	―	―	―	―
PHA-M	No	―	―	―	―
CSL3	No	―	―	―	―

*K*_D_, Equilibrium dissociation constant. ―, no affinity.

## Data Availability

The original contributions presented in this study are included in the article/Appendix A. Further inquiries will be addressed by the corresponding author.

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
