# Peer review of "Microcystis viridis NIES-102 Cyanobacteria Lectin (MVL) Interacts with SARS-CoV-2 Spike Protein Receptor Binding Domains (RBDs) via Protein–Protein Interaction"

_ijms, 2024, doi:10.3390/ijms25126696_

Round 1
Reviewer 1 Report
Comments and Suggestions for Authors
In the present study, the authors found that two lectins (MVL from Microcystis viridis NIES-102 and jacalin from Artocarpus altilis) could bind to the RBDs of the four COVID-19 strains. Their findings suggest that the natural products maybe an option for the prevention and treatment of COVID-19 infection.
1. Are there any differences of the structures or domains among MVL, jacalin and the other lectins. Are there any important or specific domains in MVL and jacalin that are responsible for the binding ability to RBD of spike protein?
2. Figure 6. Do the mutations in RBDs among the four SARS-CoV-2 strains affect the binding ability of lectins?
3. Figure 4. Does galactose inhibit the interaction between lectins and RBDs of delta and original SARS-CoV-2 strains?
Author Response
We very much appreciate the constructive comments to our manuscript “ Microcystis viridis NIES-102 algae lectin MVL interacts with the COVID-19 spike protein RBDs via protein-protein interaction.” (ijms-3048377) from the reviewers and the editors. In the following letter, the point-to-point answers for the comments will be described, in which editor/reviewer comments are in boldface. The corresponding revised parts are shown in red in the revised manuscript.
Reviewer1
In the present study, the authors found that two lectins (MVL from Microcystis viridis NIES-102 and jacalin from Artocarpus altilis) could bind to the RBDs of the four COVID-19 strains. Their findings suggest that the natural products maybe an option for the prevention and treatment of COVID-19 infection.
- Are there any differences of the structures or domains among MVL, jacalin and the other lectins. Are there any important or specific domains in MVL and jacalin that are responsible for the binding ability to RBD of spike protein?
Answer: Thank you for your comments. Regarding the differences in structures or domains among the lectins used in this study, we have summarized them in Table 1 with the addition of the "protein fold/family" entries. Thus, Table 1 is revised (page 3). MVL and jacalin do not share a common structural family and do not have specific domains except for carbohydrate recognition domain. They simply share a glycan specificity for D-galactose. The description of these is in addition to the text (page9, line270- page10, line271). An explanation of the structural differences among MVL, jacalin and the other lectins was added in the text (page 3, lines 116-118) and Table 1.
- Figure 6. Do the mutations in RBDs among the four SARS-CoV-2 strains affect the binding ability of lectins?
Answer: Thank you for your suggestion. Yes. The mutations in RBDs among the four SARS-CoV-2 strains affected the binding ability of MVL as mentioned in the text (page 5, lines 138-140). For clarity, we have added text regarding the effect of RBD mutations on lectins (page 5, lines 142-145).
- Figure 4. Does galactose inhibit the interaction between lectins and RBDs of delta and original SARS-CoV-2 strains?
Answer: Thank you for your comments. Yes. Galactose inhibited the interaction between Jacalin and RBDs of delta and original SARS-CoV-2 strains, however it did not inhibit the interaction of MVL as well as alpha and omicron. We analyzed using delta and original SARS-CoV-2 and added the data as supplemental Figure S5. We added the explanation in the text (page 7, lines 192-193; page 8, lines 206-207).
Reviewer 2 Report
Comments and Suggestions for Authors
Wang et al.; Titled “Microcystis viridis NIES-102 Algae Lectin MVL Interacts with the COVID-19 Spike Protein….”
I was shocked as soon as I started reading this manuscript. The authors apparently have missed an understanding of viral nomenclature. COVID-19 is NOT the name of the virus, it is a "disease" caused by the virus. The virus is named SARS-CoV-2. Now the authors should realize the errors they have made throughout the paper. For example, a disease cannot have a spike protein, the virus does. So, change COVID-19 to SARS-CoV-2, in throughout the whole paper. For example, change:
“S1 subunit of COVID-19” to “S1 subunit of SARS-CoV-2”;
“COVID-19 represents a class of RNA viruses” to “SARS-CoV-2 represents a class of RNA viruses”;
“The enveloped structure of COVID-19” to “The enveloped structure of SARS-CoV-2” ;
“COVID-19 spike protein RBD” to “SARS-CoV-2 spike protein RBD”, etc.
Since SARS-CoV-2 is a big name, after the first few times of using it, the authors can write “viral spike protein” etc.
Once I went past the nomenclature issue and moved into the science part, the paper looked outstanding. The experiments are logical and well done, and the conclusions justified. The techniques were state-of-the-art, particularly the carbohydrate assays and surface plasmon resonance (SPR) detection of the interaction between lectin and viral spike protein RBDs. Quantification of Kd was impressive and very methodical. The distinctly different binding behavior of the different lectins is interesting and the fine-mapping of the glycan array profiles is an exemplary study (e.g. Fig. 2 and Fig. 3).
There are many non-scientific English errors, but mainly the wrong use or absence of articles "a", "an" and also "the", almost everywhere. The paper should immensely benefit from being thoroughly checked by an English-speaking writer.
I have two minor comments/ suggestion:
The abundance of Tyr in the MVL-interaction site of the RBD (line 247) is intriguing. Some comments on its significance would befit this finding. For example, are the tyrosines located in certain places on the structure, such as on one face of the beta-strands? At least in some cases. the tyrosines may be involved in the π - π stacks, such as Y449-W38, Y489-Y4, etc.
Comments on the Quality of English Language
Mentioned in the Comments to the authors (mainly the usual problem with English articles: a, an, (the).
Author Response
We very much appreciate the constructive comments to our manuscript “ Microcystis viridis NIES-102 algae lectin MVL interacts with the COVID-19 spike protein RBDs via protein-protein interaction.” (ijms-3048377) from the reviewers and the editors. In the following letter, the point-to-point answers for the comments will be described, in which editor/reviewer comments are in boldface. The corresponding revised parts are shown in red in the revised manuscript.
Reviewer2
Comments and Suggestions for Authors
Wang et al.; Titled “Microcystis viridis NIES-102 Algae Lectin MVL Interacts with the COVID-19 Spike Protein….”
I was shocked as soon as I started reading this manuscript. The authors apparently have missed an understanding of viral nomenclature. COVID-19 is NOT the name of the virus, it is a "disease" caused by the virus. The virus is named SARS-CoV-2. Now the authors should realize the errors they have made throughout the paper. For example, a disease cannot have a spike protein, the virus does. So, change COVID-19 to SARS-CoV-2, in throughout the whole paper. For example, change:
“S1 subunit of COVID-19” to “S1 subunit of SARS-CoV-2”;
“COVID-19 represents a class of RNA viruses” to “SARS-CoV-2 represents a class of RNA viruses”;
“The enveloped structure of COVID-19” to “The enveloped structure of SARS-CoV-2” ;
“COVID-19 spike protein RBD” to “SARS-CoV-2 spike protein RBD”, etc.
Since SARS-CoV-2 is a big name, after the first few times of using it, the authors can write “viral spike protein” etc.
Answer: Thank you for your suggestion. As the reviewer pointed out, we used the wrong word "Covid-19" for SARS-Cov-2. The error has been corrected according to the reviewer's comments and the English-speaking writer's review including the title. So the title is changed.
Once I went past the nomenclature issue and moved into the science part, the paper looked outstanding. The experiments are logical and well done, and the conclusions justified. The techniques were state-of-the-art, particularly the carbohydrate assays and surface plasmon resonance (SPR) detection of the interaction between lectin and viral spike protein RBDs. Quantification of Kd was impressive and very methodical. The distinctly different binding behavior of the different lectins is interesting and the fine-mapping of the glycan array profiles is an exemplary study (e.g. Fig. 2 and Fig. 3).
Answer: Thank you for your kind comments.
There are many non-scientific English errors, but mainly the wrong use or absence of articles "a", "an" and also "the", almost everywhere. The paper should immensely benefit from being thoroughly checked by an English-speaking writer.
Answer: The English-speaking writer has reviewed the manuscript and carefully revised the text according to the English proofreading.
I have two minor comments/ suggestion:
The abundance of Tyr in the MVL-interaction site of the RBD (line 247) is intriguing. Some comments on its significance would befit this finding. For example, are the tyrosines located in certain places on the structure, such as on one face of the beta-strands? At least in some cases. the tyrosines may be involved in the π - π stacks, such as Y449-W38, Y489-Y4, etc.
Answer: Thank you for your useful comments. Your suggestion gave us a great insight, and we checked the in silico docking results again carefully, but unfortunately the Tyr residue is not in some specific positions such as on the α-helix or β-strand. On the other hand, as you mentioned, even though Tyr not located on special structures, Tyr and Trp residues are actively involved in pi-pi stacking such as Y449-W38, Y489-Y4, and other Tyr residues also contribute to the interaction of lectins with RBDs in the form of H∙∙∙π interactions. We added the discussion in the text (page 10, lines 294-302).
Comments on the Quality of English Language
Mentioned in the Comments to the authors (mainly the usual problem with English articles: a, an, (the).
Answer: As mentioned above, the English-speaking writer has reviewed the manuscript and carefully revised the text according to the English proofreading.